# Sustainable Management of Very Large Trees with the Use of Acoustic Tomography

**Margot Dudkiewicz** [1,*] and **Wojciech Durlak** [2]

1    Department of Landscape Architecture, Faculty of Horticulture and Landscape Architecture, University of Life Sciences in Lublin, Głęboka St. 28, 20-612 Lublin, Poland
2    Horticultural Production Institute, Faculty of Horticulture and Landscape Architecture, University of Life Sciences in Lublin, Głęboka St. 28, 20-612 Lublin, Poland; wdurlak@autograf.pl
*    Correspondence: margotdudkiewicz@o2.pl; Tel.: +48-696-900-453

**Abstract:** This manuscript summarizes the process by which large trees are protected in Poland, how they gain protected status, and the use of acoustic tomography to assess the structural soundness of five individual trees. The authors discuss tree stressors and tree care options, and how the outcome of each assessment was used in the decision-making process. Moreover, the use of acoustic tomography as an assessment tool helps to gain public acceptance for the assessment. In sustainable development, there is talk of the conscious management of urban vegetation, and specifically tree populations in the city. In line with sustainable development, actions are taken to manage the existing natural resources, e.g., historic trees, properly. Thanks to using an acoustic tomograph, it is possible to diagnose old tree specimens, take care of the safety of people and property near the natural monument, and to test candidates for their eligibility as monuments. Thanks to the research presented, it was ordered that one poplar tree be left and observed in Lublin, that two linden trees be cut down in Sandomierz and Lublin, that arborist works consisting of lowering the height of a linden tree in Sandomierz be undertaken, and that monument protection be applied for an ash tree in Łęczna. A visual tree assessment (VTA) was the starting point for this research. Each of the trees could endanger the safety of site users, and the diagnostics performed using sound waves were crucial in assessing their health condition. Our results highlight that acoustic tomography is an essential diagnostic method applicable to trees belonging to cultural heritage, such as old trees, and is effective in preventive tree management through the monitoring of mid- to long-term changes in internal decay or cavities that are difficult to diagnose with the naked eye.

**Keywords:** sustainable management of urban greenery; historical greenery; urban greenery; natural monument; very large trees; nondestructive assessment of tree; PiCUS sonic tomograph 3; acoustic tomography

## 1. Introduction

It is optimal for a city's development to obtain economic and ecological benefits at the same time. Finding a balance between the social, ecological, and economic aspects is one of the biggest challenges in sustainable city management [1]. Shaping properly functioning urban green areas with large, historic trees preserved to enrich the urbanized environment may be one way to build this balance. Moreover, there is a need to properly shape and protect the urban landscape in order to create an image of the city as more friendly and harmonious [2–4].

Green areas in the city fulfill various natural (e.g., climatic, hydrological, biological) and non-natural functions (e.g., social, recreational, aesthetic). Public green areas include, among others: parks, squares, the street landscape and city forests. In the composition of the city, greenery structures the layout; it connects, divides, separates, isolates, masks, and adorns. City inhabitants want to live among greenery, which is confirmed by research and urban policy [5–11]. Green areas are the essential components of the natural environment

and the urban landscape, as the organic material that binds together the urban fabric. The elements and forms of greenery, which are an integral part of a city's landscape, contribute to its identity. In the spatial structure of urban greenery, apart from large objects, such as parks, one should also consider the preserved natural and historical elements, which include veteran trees. Historical greenery management involves the conservation and care of historic buildings/areas. Conservation is the proper maintenance and activities aimed at securing and preserving the substance of very large trees and stopping the processes of their destruction. Considering historic trees, continuous, ongoing care, and keeping them in good condition while maintaining the appropriate conservation standards is particularly important [12–14]. In urban areas, there are occasions in which individual trees can fail under the influence of high wind loads, thus, causing a danger to people and property [15]. A large part of this urban greenery comprises thousands of mature trees that can be found growing along roadsides, in parks, and within residential, industrial, shopping, and business districts. This research aimed to verify the usefulness of the PiCUS 3 sonic tomograph in assessing the health condition of trees with monumental dimensions for the sustainable management of urban vegetation, specifically the tree population. The use of this technology consists of an advanced level inspection after a visual and basic inspection. Until now, invasive techniques have been used to assess the health condition of trees. The previously used tools, i.e., penetrometers, gauges, drills or electric sensors that require drilling measurements, are still used, although to a lesser extent. Undoubtedly, the decline in the popularity of these methods was largely due to the fact that the tests left wounds, which often weakened a tree's vitality. The authors present the principles of the protection process of large trees in Poland, the method of obtaining protected status for them and the use of acoustic tomography to assess the structural strength of five individual trees. Tree stressors and tree care options are discussed, as well as how the results of each assessment were used in the decision-making process.

Old trees are protected not only as natural forms but also as cultural objects. The protection motives are also for scientific, genetic, landscape, ecological, historical (memorial), patriotic or didactic purposes [16,17]. Veteran trees have witnessed the history of many cities. They are not only picturesque elements of the cultural landscape (often painted, photographed and described), but are also an extremely important element of the natural world. They are also very important refuges of biodiversity with sometimes rich accompanying flora. Many species of lichens live on tree trunks, including rare and protected ones. Branches of trees are a convenient place to set up bird nests, and in the treetops you can meet many species of birds (including legally protected ones) such as nightingales, mazurkas, blackbirds, bunting, kestrels, finches, and squeaks. In cities, apart from the ecological and aesthetic aspects, the psychological impact of very large trees on society as a natural remnant of nature is essential. These objects in an artificial environment, including urban areas, are an element that humanize the space [18]. These monuments of nature, in addition to their natural and historical values, have a considerable tourist value, as evidenced by, inter alia, the examples of tourist routes that are built on their attractiveness. An important aspect of this is developing appropriate attitudes towards valuable components of the environment, both among residents and visitors. It can be achieved through adequately prepared tourist offerings explaining the historical significance of natural sites [19–21].

*Protection of Natural Heritage Resources in Poland—Natural Monuments*

According to the Central Statistical Office (2018), trees constitute the most numerous group of all nature monuments–94.7%. These largely include single trees, but also their groups, alleys, and shrubs [22].

In Poland several categories have been established for classifying trees as natural monuments: *a single tree* as a natural monument, *groups of trees*—where several trees constitute one natural monument, and *avenues*—where several dozen or even several hundred trees form one natural monument [23]. The premise for a tree to be protected as

a monument is usually based on its circumference at the breast height. The dimensions that qualify a tree or shrub to be considered a natural monument vary and are smaller for shrubs and native trees, and larger for exotic trees. In the annex to the regulation, we found the minimum circumference of trunks (measured at a height of 130 cm) for individual types and species of trees including, e.g., ash trees at 250 cm, linden trees at 300 cm, while for magnolia or ginkgo (which is rare in Poland), it is only 150 cm [24,25]. Among the over 35,000 entries, there are 29,500 entries in the register of all types of natural monuments that are individual trees, 3500 groups of trees, and approximately 700 avenues. The monumental trees show a great variety of species; currently they represent 71 genera with 171 species, including 40 conifers and 131 deciduous, of which 60 are native and 111 introduced (foreign origin). The species diversity of the trees newly established as monuments is increasing. Most of the monumental conifers are pines, larches, junipers, spruces, yews, and Douglas firs. Most of the monumental deciduous trees are linden, oak, maple, beech, chestnut, ash, and hornbeam. Linden and oak trees constitute two thirds of all monument trees in Poland [21,24], but the number of natural monuments is not constant.

Care of the veteran trees includes cutting to remove deadwood branches, cuts to their crowns limiting their weight and volume, as well as the installation of bindings and elastic protections. Conservation also includes the management of natural monuments and a routine inspection in detail should be carried out not only through visual assessment but also through monitoring using modern technology [26,27].

Abolition of the monument protection of trees should always be preceded by a detailed assessment of the tree's health condition and the threat that the tree causes. An inventory should be made of the presence of protected species, and an analysis of the possibility of performing treatments that eliminate or minimize the threat.

It is worth mentioning that dead trees should still be preserved if they do not threaten safety because they have an exceptional natural and historical value. This can be done in various ways:

- Replacement, planting near a young tree—e.g., the "Copernicus oak" in the courtyard of the Cathedral in Frombork,
- Conservation of tree trunk—e.g., the elm in front of Pawiak in Warsaw. After decay, the stump was preserved for many years, and then a bronze monument was made,
- Using parts of a rotten trunk—as sculptures,
- Leaving the tree standing, for example, as the frame for creepers,
- Leaving the tree lying down,
- Ex situ protection of remains—transfer of trunk fragments for a museum, e.g., with the dates of historical events marked on a cross-section.

## 2. Materials and Methods

The numerical parameters characterizing the objects were determined as the circumference at the height of 130 cm, crown reach (m) and tree height (m). The height of the trees was measured with a Nikon Forestry Pro laser rangefinder. The range of the crown was measured with a Leica rangefinder. The most protruding branches were taken into account and two measurements were read, west to east and north to south, then the sum was divided into two.

The analyses included five trees of considerable size, growing in selected cities in eastern Poland, in Lublin, Sandomierz and Łęczna. The trees were: a black poplar (*Populus nigra* L.) at 3 Weteranów Street in Lublin; a black poplar (*Populus nigra* L.) in Litewski Square in Lublin; a small-leaved lime (*Tilia cordata* Mill.) at 67 Kwiatkowski Street in Sandomierz, a small-leaved lime (*Tilia cordata* Mill.) at 3 Staromiejska Street in Sandomierz and a European ash (*Fraxinus excelsior* L.) growing at Marszałek Piłsudski Street in Łęczna. The studies were conducted during 2017–2020. Each of the specimens was a different case, extremely difficult to assess. The trees had previously been VTA-examined by other arborists and the opinions included only tree assessments using the VTA method. The

following tests using acoustic tomography were the first to be performed with the use of specialized tools. Each of the trees could endanger the safety of site users, and the diagnostics performed using sound waves were crucial and final in assessing their health conditions.

Visual tree assessment (VTA) is a non-invasive method of examining the health and structural condition of individual trees. The assessment provides information on the condition of the roots, trunk, main branch structure, crown, buds and leaves together providing an assessment of general tree health and vitality. The obtained observation results are then compiled into special forms.

The trees were assessed using the PiCUS 3 sonic tomograph of the German company ARGUS GmbH. Based on the obtained results generated by specialized software in the form of tomograms, decisions were made resulting in either felling, leaving a tree in its current state with appropriate conservation recommendations, or by applying for its legal protection. The device uses the speed of sound waves in the wood which depends on its density and flexibility. The device records the time of the acoustic waves using a system of sensors, generated at each measure point. Most defects inside the trunk, especially the presence of decay, lead to a reduction in the density and flexibility of the wood, which results in a reduction in the speed of waves at the point of damage. The results of the conducted research depict a tomogram, which is a graphic picture that presents the actual internal structure of a trunk cross-section at the height of a tree examination in detail and in some cases allows us to detect the early stages of tree decay.

The analysis began with tapping the trunk with a rubber hammer in order to determine specific sound effects, which allowed us to determine the level of the measurement points. Then, at the designated height, measurement pins were placed to attach the sensors responsible for receiving the sound waves. Usually, 5 to 24 measuring points are assumed to be spaced apart by certain values. The number of points placed depended on the circumference and shape of the tree trunk (Figure 1). Measurements were made at the height of 130 cm.

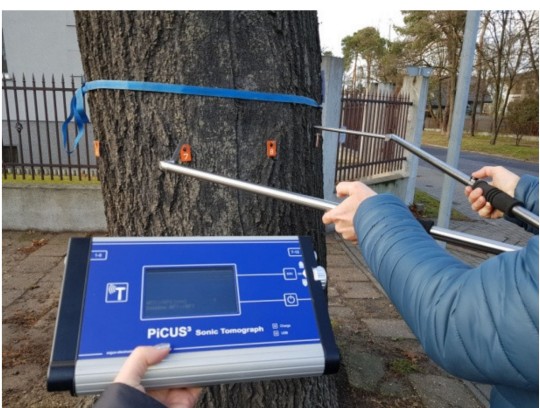

**Figure 1.** CT sonic scanner located on a tree trunk (M. Dudkiewicz, 2021).

The next step was to measure the tree trunk geometry, which was assessed at the measurement level. In order to accurately reproduce its shape, an electronic caliper (PiCUS Calliper) was used, after which the measurement data were wirelessly transmitted to the central unit of the tomograph, where they were read on the screen (Figure 2).

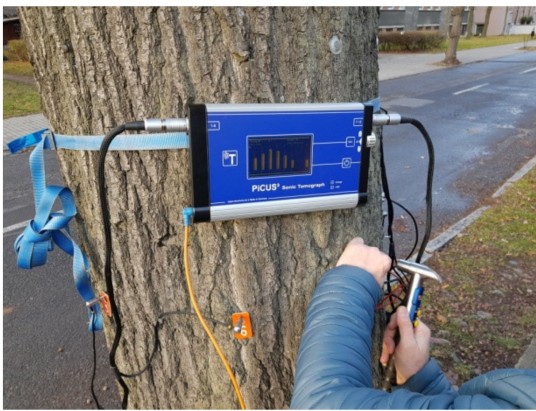

**Figure 2.** Measuring the geometry of the tree trunk using the PiCUS Calliper (M. Dudkiewicz, 2021).

Then, an acoustic measurement was made using a radio hammer. With its help, a sound pulse was generated at each subsequent measuring point, which was received by the sensors. During the generation of the wave, sensors recorded the time of receiving the signal. This stage was crucial because a visualization was then created, showing the inside of the trunk (tomogram) with the exact size and location of the defect at a given height. The stage of the acoustic diagnosis resembles the work of a woodpecker, hence the name. The image shown on a tomogram is usually relatively easy to interpret. Shades of brown indicate sound wood, green areas indicate early decay, violet is decayed wood and blue indicates a cavity, a crack or wood so decayed that it hardly transmits any sound waves. Using the images gathered with a tomograph, the critical areas for further testing can be identified. The yellow lines on the cross-section of the trunk suggest the appearance of internal cracks, which are extremely dangerous because often on the outside there are no obvious visible symptoms that could confirm this. Thus, these are the weakest areas of the wood structure with progressive decay and destruction of wood tissue. An additional advantage is the possibility of calculating the so-called coefficient determining the mechanical strength of the trunk (t/R), i.e., the ratio of healthy wood (t) to the tree trunk radius (R). According to the available literature, the t/R ratio should not be lower than 0.33, and in the case of trees with closed cavities, even 0.3 [28–31].

## 3. Results

### 3.1. Objects and Reasons for Tomograph Examination

The research started with the baseline dendrological measurements (Table 1).

**Table 1.** Data from inventoried trees (authors).

| No. | Species Name | GPS Location | The Circumference of the Trunk at the Height of 1.3 m (cm) | Height (m) | Crown Reach (m) |
|-----|--------------|--------------|---------------------------------|------------|-----------------|
| 1 | Black poplar (*Populus nigra* L.) | 51°144′53.387″ N 22°322′16.2166″ E | 476 | 21.8 | 17 |
| 2 | Black poplar (*Populus nigra* L.) | 51°144′52.999″ N 22°333′33.466″ E | 506 | 16 | 7 |
| 3 | Small-leaved lime (*Tilia cordata* Mill.) | 50°411′50.76″ N 21°444′15.122″ E | 577 | 19.7 | 15 |
| 4 | Small-leaved lime (*Tilia cordata* Mill.) | 50°40′37.89″ N 21°44′39.377″ E | 520 | 8.4 | 9.45 |
| 5 | Ash (*Fraxinus excelsior* L.) | 51°18′7.90″ N 22°53′11.722″ E | 470 | 30.7 | 18.5 |

**Object No. 1—black poplar (*Populus nigra* L.)** (Figure 3)

**Examination reason:** the fear of parents of children playing in the playground about the poor condition of the tree and the potential risk of it breaking.

Near the tree there is a playground on the north side and garages on the south side. This poplar is a valuable natural and historical element of the estate due to its size and relatively good condition, despite various anomalies that occur on the trunk or in the tree's crown. It is noticeable that the conservation works carried out properly over the years contributed to preserving the tree in a decent condition. Nevertheless, over time a tree slowly undergoes destructive processes, mainly in the inner part of the trunk. The performed tomographic examination of this tree showed an obviously progressive process of caries. The scope of the most significant damage covers about 27% of the entire trunk cross-section, shown in blue and purple colors on the tomogram (Figure 4). On the other hand, technically sound wood covers an area of 62% of the trunk's cross-section. The green color (the remaining percentage of the trunk cross-section) is the so-called transition wood characterized by a slightly weakened structure but not yet fully damaged. It indicates which direction the infection may continue to spread. The calculated t/R coefficient of the mechanical strength of the trunk at the measurement height was 0.29, and its value was slightly lower than that considered to be safe. Anything below this limit may increase the risk of the tree trunk breaking at the least expected moment. In conclusion, it was recommended to leave the tree in its current state with regular observation and monitoring of the pace of its developing destructive processes.

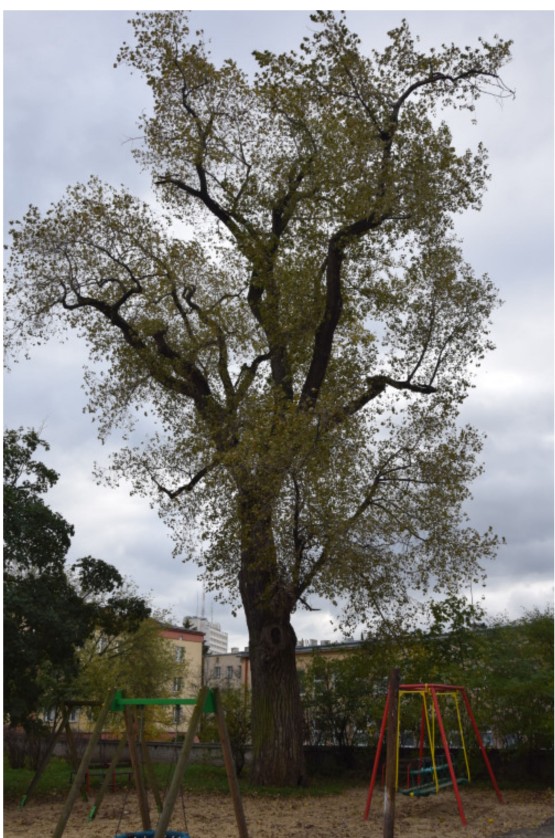

**Figure 3.** Black poplar at Veterans St. in Lublin—general view of the tree (photo by W. Durlak, 2017).

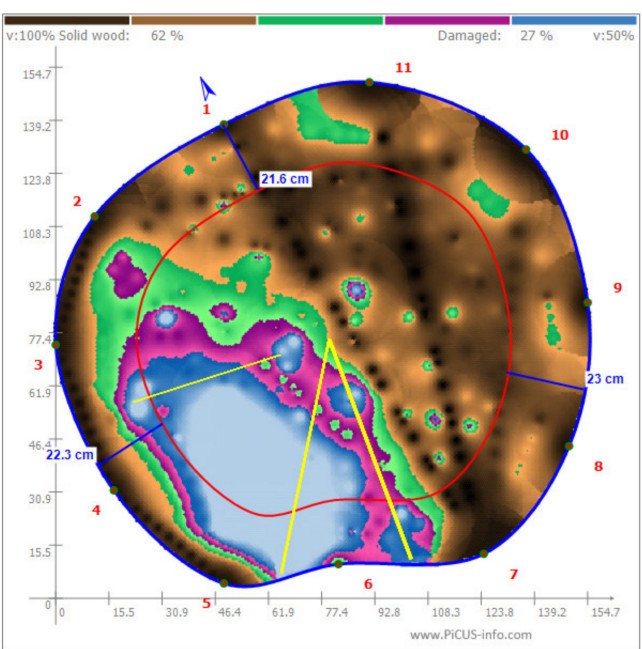

**Figure 4.** Tomogram of the inside of the black poplar trunk inv. No. 1.

**Object No. 2—black poplar (*Populus nigra* L.)** (Figure 5)—a natural monument, a tree essential to the inhabitants, and a meeting place for many generations of Lublin residents.

**Examination reason:** commissioned by the City Hall and Provincial Conservator of Monuments in Lublin, the square was rebuilt in 2016, during which the poor health condition of the tree was revealed (VTA method).

The poplar was planted around 1880 after the January Uprising as the so-called "Tree of Freedom". In the 1940s, the tree's roots spread over the surrounding lawns. About 20 years later, pavements were laid near the poplars which threatened the soil conditions. The condition of the tree deteriorated in the early 1990s when part of the root system was cut off during construction works. In the early 2000s the tree began to decline, possibly due to insufficient hydration. In 2008, an underground irrigation and aeration system was installed with the purpose of improving root function. In 2013, the poplar was approximately 16 m high. In December 2016 *Pleurotus ostreatus* appeared and was likely a symptom of decay. During the research, the visual condition of the tree was unsatisfactory. The crown was very asymmetrical and significantly lowered due to the care works carried out over the years. A fragment of the leading guide and three branches remained in its upper part. From the south, the tree trunk was devoid of bark to the top. After the tomographic examination, it was found that a very advanced process of decomposition of the wood tissues was developing inside the trunk (Figure 6). The technically efficient wood occupied only 15% and damaged wood 71% of the trunk cross-section. The remaining 14% was occupied by transitional wood. The advanced process of decomposition was evidenced by the light blue color shown on the tomogram, covering a vast area of the inside of the tree trunk. The limiting wall thickness, which allowed for the determining of the minimum mechanical strength of the tree trunk, was calculated at an average of 23.2 cm. Still, at no point on the trunk cross-section was such a parameter met. The calculated t/R ratio was 0.28. The occurrence of fungi causing extensive internal destruction of the trunk contributed to a significant weakening of the mechanical strength of the trunk, which ultimately determined the fate of the tree. Despite enormous protests from the inhabitants, the tree was intended to be felled for health reasons and the surrounding area's safety. A new poplar was planted in its place. The research results were presented at a press conference, which contributed to a peaceful end to the dispute over the felling of trees between the Municipal Office and the Provincial Office in Lublin and the inhabitants of Lublin.

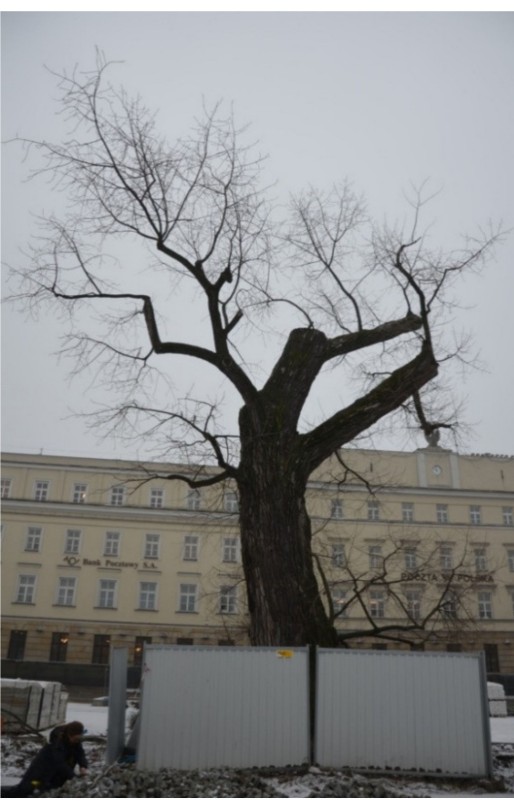

**Figure 5.** Black poplar in Litewski Square in Lublin—general view of the tree (photo by M. Dud-kiewicz, 2016).

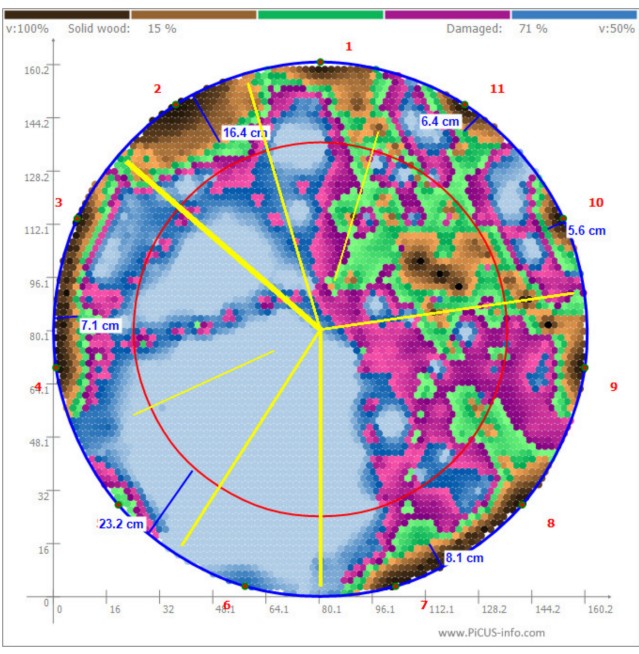

**Figure 6.** Tomogram of the inside of the black poplar trunk inv. No. 2 (by W. Durlak, 2016).

**Object No. 3—small-leaved lime (*Tilia cordata* Mill.)** (Figure 7)—a natural monument.

**Examination reason:** the fear of the City Hall in Sandomierz and the property owner about the poor condition of the tree, the planned construction of a shop, and a car park near the tree.

The tree was located near the newly built store. In terms of accessibility, the tree was in an average condition with a trunk with a large cavity. The tree was hit by lightning several

times. The processes of tissue decomposition in the examined tree were advanced and covered practically the entire cross-section of the trunk. (Figure 8). The wood in working order occupied only 27% of the whole trunk cross-section and damaged wood 61%. The remaining area was so-called transitional wood, with a slightly weakened structure and taking up 12% of the trunk cross-section. The attached tomogram shows the progressive destruction of the central structures of the trunk spreading towards the edge in all directions (light blue and purple). On the eastern and south-eastern sides, it was most advanced. Many factors could have contributed to this state of affairs, with extreme damage to the trunk and a developed fungal infection combined with the breaking of one of the main branches (from the southeast side). As a result of enormous wounds, the infection and tissue decomposition process was significantly accelerated, significantly weakening the mechanical strength of the trunk. The limit value marked on the tomogram with a red line, determining the minimum mechanical strength of the trunk, was on average 27.4 cm. The t/R ratio calculated on its basis was 0.29. After taking the measurements, the visual assumptions about the health of the tree were confirmed. Unfortunately, the situation qualified the tree for deletion from the register of natural monuments and removal due to the excessive threat to the safety of potential users of the site. At the site of the growing tree, it was suggested that the topped trunk should be left as a witness to the story. In this case, the discussion at a session of the Sandomierz City Council and the publication of the results of the research made it possible to silence the disputes about the legitimacy of tree felling.

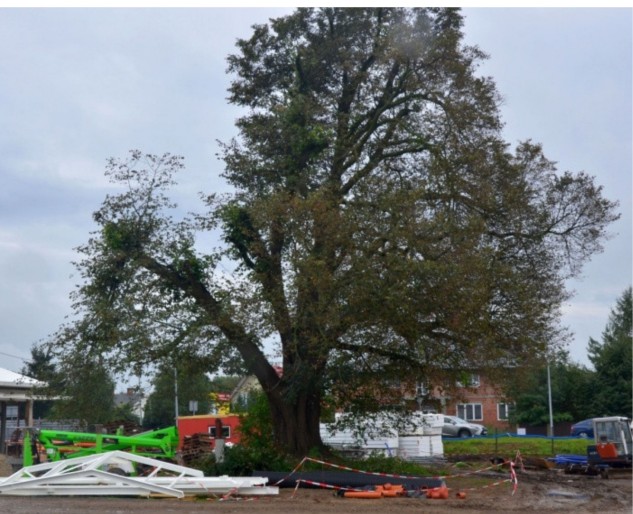

**Figure 7.** Small-leaved lime at E. Kwiatkowski St. in Sandomierz—general view of the tree (photo by M. Dudkiewicz, 2018).

**Object No. 4—small-leaved lime (*Tilia cordata* Mill.)** (Figure 9)—a natural monument.

**Examination reason:** the fear of the City Hall in Sandomierz and the owners of the property, i.e., the Dominican Monastery Complex, about the poor condition of the tree since the tree grows at the main entrance to the church.

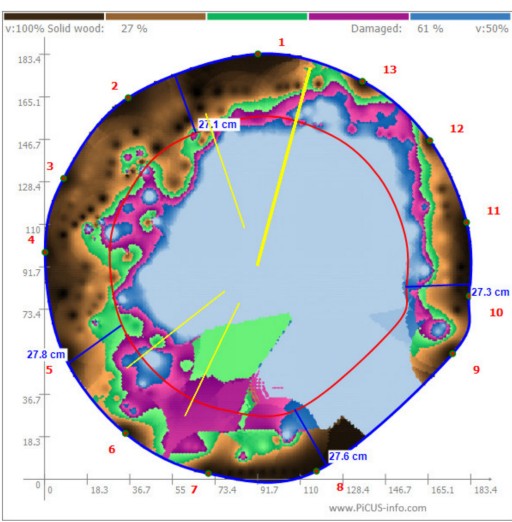

**Figure 8.** Tomogram of the inside of the small-leaved lime tree trunk inv. No. 3.

The specimen grows near the communication route leading to the church. The distance between the tree and the temple is 11 m. The visual analysis combined with the interpretation of the results obtained with the acoustic tomograph revealed a rapidly progressive degradation of the tissues inside the trunk (Figure 10). In the case of this tree, it was 0.17. According to the available literature, this coefficient should not be lower than 0.33 and in the case of trees with closed cavities even 0.3 [28,29,31]. All values below this limit significantly increase the risk of a tree breaking or a trunk splitting at the least expected moment. This risk increases even more when early signs of cracking, fungal infection or very thin walls appear inside the trunk, as is the case here. In the attached tomogram between the above-mentioned measurement points (on the west side and on the east side) a safe wall thickness simply does not exist. A large proportion of the boughs in the upper part of the crown had recently been removed due to rapidly progressing decay. The remaining bough fragments generated an excessive load on the rest of the crown, which had recently resulted in numerous fractures in large sections, mainly at the point of forking. It was a significant threat to potential users of the area. A decision was made to lower the crown to the place of outgrowths in the eastern part of the tree's crown, just above the chapel embedded in the trunk, which allowed for significantly relieving the tree's trunk and reducing the trunk torsional forces acting on the tree. There was also a band bonding the trunk, which prevented its further breaking. If despite everything the applied measures do not improve the safety of the tree itself and in its immediate vicinity, the only sensible solution will be to remove it after deleting it from the register of monument trees, but in such a way that a part of the trunk remains in its place, giving evidence of the existence of a historical object which is this monumental, small-leaved lime. A good solution may be to plant a new specimen inside the old stump after removing as much of the muck and infected soil from its interior as possible, decontaminating its remains, and supplementing with a layer of a new substrate with the use of mycorrhizal vaccines. Such a procedure requires lowering the remains of the trunk so that the new tree has good conditions for growth and development.

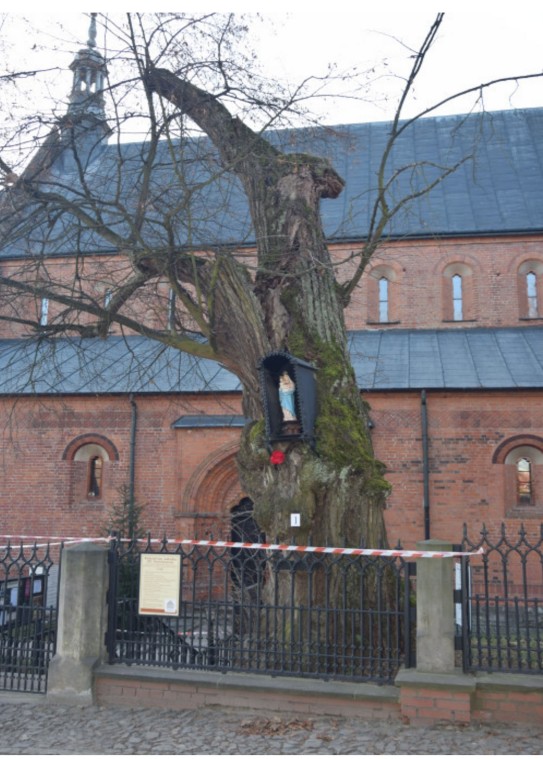

**Figure 9.** Small-leaved lime tree at 3 Staromiejska St. in Sandomierz—a general view of the tree from the north (photo by M. Dudkiewicz, 2020).

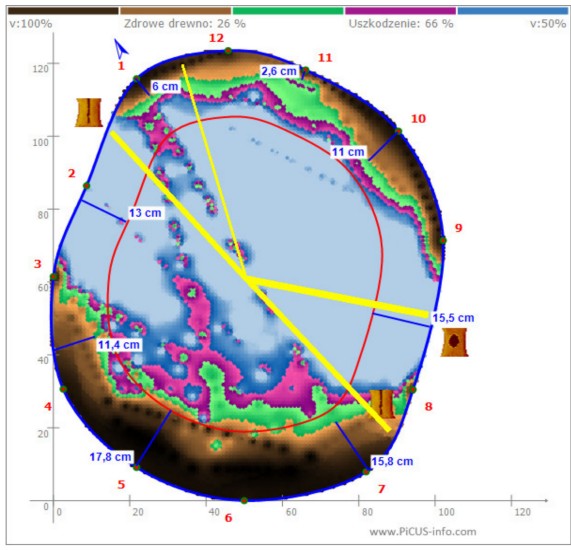

**Figure 10.** Tomogram of the inside of the small-leaved lime tree trunk inv. No. 4.

**Object No. 5—European ash (*Fraxinus excelsior* L.)** (Figure 11).

**Examination reason:** request of local social activists for a dendrological examination of the tree to cover the specimen with legal protection.

The tree grows at one of the main communication arteries in Łęczna (small city near Lublin)—Marszałka Piłsudskiego Street. The tree has a very symmetrical crown. At a height of 3.5 m, the trunk forks into two trunks, while the bifurcation is U-shaped. After the tomographic examination and analysis of the results, slight damage to the core part of the tree trunk was found (Figure 12). The technically sound (healthy) wood occupied 85% of the trunk cross-section, which is suggested by the brown color of various shades. Damaged

wood, in turn, covered 8% of the trunk cross-section and temporary wood the remaining 7% of the examined area. The geometric moment of inertia calculated for different directions of the trunk cross-section, measured at the measurement height, amounted to 92.6 to 97.7% of the maximum strength for the trunk without defects or damage. Based on the obtained results, it was possible to conclude that the tree was relatively good in terms of its mechanical strength and resistance to trunk bending. The minimum remaining wall thickness for this tree as a safety limit against trunk fracture was, on average, 21.7 cm. Therefore, the tree had a huge supply of healthy tissue, making it safe for the environment. In this particular case, the specimen was in excellent health, despite its advanced age. This tree is the last remaining of such a large specimen from the old alley leading from the Church of St. Maria Magdalena in the direction of the cemetery at John Paul II Street. Due to this fact, the protection and care of this tree are significant not only for historical but also for landscape and social reasons. The conducted research and the obtained results provide the basis for covering the assessed tree with legal protection and entering it into the register of very large trees. This will allow for the adequate protection of the last of the specimens included in the former avenue system and increase the area's prestige where the tree grows. Such an initiative is necessary because there is a well-founded fear that this tree will share the fate of the ash trees that have been cut down in the alley in recent years. The proposed changes were finalized at a session of the Commune Council, where in 2021 based on a resolution, the tree was entered into the register of very large trees.

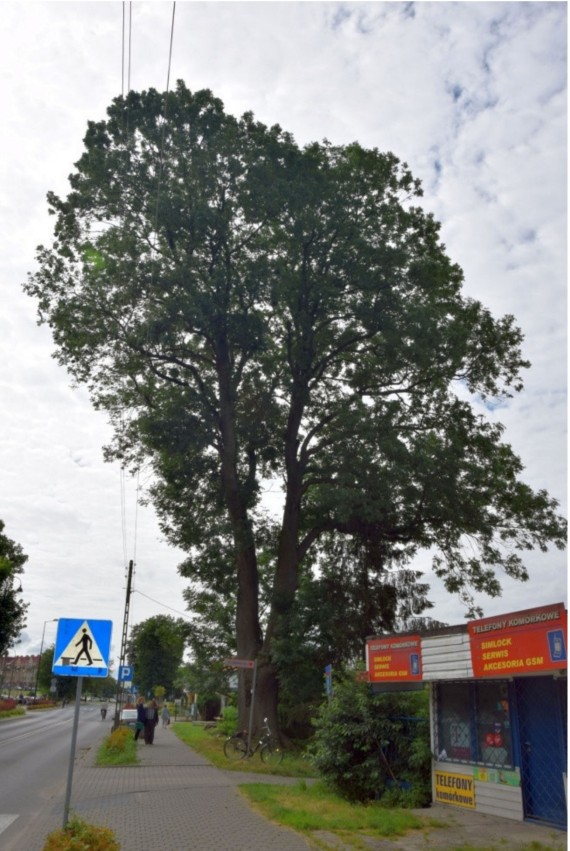

**Figure 11.** The ash tree at Marszłka Piłsudskiego St. in Łęczna—general view of the tree habit (photo by M. Dudkiewicz, 2021).

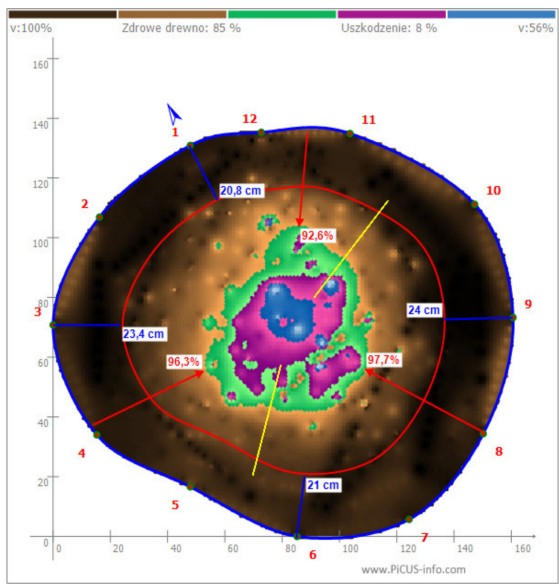

**Figure 12.** Tomogram of the interior of the common ash trunk inv. No. 5.

*3.2. Characteristics of Threats to Very Large Trees*

As a result of the above research, certain regularities were noticed. Trees growing in urban conditions are exposed to various types of stress. The most significant stressful factors of the urban environment related to communication arteries include the following abiotic factors: high temperature, soil drought, oxygen deficiency, excess of heavy metals, soil salinity, nutrient deficiency, alkalization or acidification of the substrate, ice or snow cover, low or excessive tree density, and lack or excess of soil microorganisms. Abiotic factors are industrial and automotive pollution, soil compaction, and electromagnetic fields. In the case of the above specimens, the effects of various stresses were often cumulative (Table 2).

**Table 2.** Results of dendrological expertise and factors damaging very large trees (by authors).

| No. | Species Name | t/R Ratio | Damaged Wood | Recommendation | Causes of Destruction Likely |
|---|---|---|---|---|---|
| 1 | Black poplar (*Populus nigra* L.) | 0.29 | 27% | monitoring and cutting in the future | age, soil pollution, air pollution |
| 2 | Black poplar (*Populus nigra* L.) | 0.28 | 71% | cutting | age, soil pollution, air pollution, groundwater level change |
| 3 | Small-leaved lime (*Tilia cordata* Mill.) | 0.29 | 61% | cutting | age, soil pollution, air pollution |
| 4 | Small-leaved lime (*Tilia cordata* Mill.) | 0.17 | 66% | lowering the crown | age, soil pollution, air pollution |
| 5 | Ash (*Fraxinus excelsior* L.) | 0.3 | 8% | preservation and under protection | age, location near one of the main communication arteries of the city (significant soil and air pollution) |

**4. Discussion—Tomograph as a Tool Supporting the Management of Historical Greenery**

The safety in historic parks and avenues with large trees is vital and requires precise diagnostic techniques to detect decay and other types of structural defects of tree trunks. Visual tree assessment (VTA) is still the starting point for such studies [32–34]; however, the internal defects of tree trunks often remain beyond the sight of an arborist or a botanist [35]. For many years, the only available instrument for a detailed assessment of the inner wood structure of a growing tree was the Pressler drill; however, this method requires

intervention in the internal tissues of the tree. Its use in the case of precious historic trees is controversial [36–38]. The most appropriate methods minimize the destructive impact of the research on the tree, such as acoustic tomography using PiCUS tools [36]. For more precise results, the use of acoustic tomography in combination with electrical resistance tomography is suggested. As both research methods complement each other, a detailed image of the inside of the tree trunk can be obtained [39]. It is worth noting that the electric tomograph will not work with dry wood.

Compared to other methods, acoustic tomography is very effective even in the early stages of wood decomposition [40]. Unlike other instruments used to detect trunk anomalies, an acoustic tomograph does not require drilling or breaking the barrier created by the tree to limit or slow down the spread of decay. New diagnostic methods have been developed to detect infections and decay of internal trunk structures using acoustic or electrical tomography in recent years.

Acoustic measurements have been used for strength classification of utility poles since the 1960s. Tapping with a hammer on one side of the pole induces a sound wave. A sensor at the other end of the pole detects when the stress wave arrives. The distance between the tapping point and detector location is then divided by the measured time of flight, delivering a sonic speed. The higher this speed, the higher the modulus of elasticity, which is a good measure of wood quality and load carrying capacity. Starting in 1990, this principle was used on standing trees in order to identify internal damages [41,42]. This is a well-established tool for the assessment of a tree's structural soundness, with numerous studies over the past 20 years testing its application under various conditions. Many research teams have confirmed the effectiveness of acoustic tomography in detecting tree decay [43–50]. At the beginning of the 21st century, research was conducted on the health condition of urban trees using various research methods (electrical and acoustic tomography and GPR), achieving various degrees of success with them. Among the methods used, acoustic tomography turned out to be the most effective nondestructive tool for detecting internal tissue decomposition, which, according to the authors, is the most accurate in locating anomalies and estimating their dimensions and shapes. Frank Rinn, the inventor of impulse tomograph, is world famous as an instructor on tree inspection. Frank Rinn invented and developed different methods, machines, and computer programs for various applications in tree-ring analysis [51,52]. The basics, potential uses and limitations of the resistance-recording drilling method for tree inspection and risk management are some examples of what F. Rinn has presented in many of his works. Research has confirmed that in addition to technical specifications, an understanding of wood anatomy and mechanical material properties is required for the proper application of this method and reliable interpretation of results [53,54]. The principle of operation of the resistor is based on the measurement of changes in the resistance generated when drilling wood. The densities of healthy wood and wood decomposed by fungi differ, therefore, on the tested cross-section of the tree, zones of wood with reduced strength and possible total losses can be indicated. The examination allows for identifying the presence of soft rot and to determine its size, and is included in the dendrological expertise [54].

Pulse tomography can also be used to manage and protect the floodplain deciduous forest. The results of the simulation and growth model supplemented with wood tomography—both non-invasive methods—made it possible to adjust the management plan for protected areas in the Czech Republic [55]. PiCUS Sonic is the first tomograph acoustic used for the calculation of the relative velocity values of sounds. Earlier devices, based on absolute speed values (which differ between species of trees, between trees of the same species, and even within the same trees), were only detectable in large damage by comparing the speeds obtained from the measurement with tabular values. In the case of the PiCUS Sonic, there is automatically a conversion of all measured values' speeds at "relative" speeds, giving more opportunities to diagnose correctly the damage inside the trunk. Gilbert and Smiley (2004) rated the PiCUS Sonic tomography accuracy at 89% and confirmed that the rapid prevention of degradation of the protected specimen and arbori-

cultural work to maintain the tree in a proper condition (e.g., proper pruning, lowering the crown), contributes to preserving the rich biodiversity of the city [43].

One of the latest studies presented, investigated the reliability of non-invasive measurements using acoustic tomography to detect internal defects in *Abies holophylla* Maxim. trees and compared the results with measurements using the invasive method of resistance micro-drilling. The tomograms were visually compared with tree cross-section images. The results of acoustic tomography and resistograph measurements showed no significant differences, while the explanatory power, as determined by a regression analysis, were considerably high at 67% with a positive correlation between the two methods. In comparison to the cross-section images, the tomograms were found to reflect the size and position of internal decay, although the detected size tended to be larger than the actual decay area. These studies indicate acoustic tomography is a promising non-invasive technique for detecting internal defects in very large trees [56].

The authors of this manuscript conducted similar research (as one of the first in Poland) in the case of the historic greenery of the Botanical Garden in Lublin. It was possible to confirm the preliminary assumption that sound tomography can be a critical element in the sustainable management of historical greenery in botanical gardens [57].

Consideration of tree hazard risks can inform broader decisions on long-term vegetation management to enable a functioning, healthy, and safe urban environment for future users of roads and recreation areas. Assessing failure potential requires a recognition of weaknesses, of the conditions that act on those weakness, and of the frequency at which those conditions occur. Failure is the end result of many interacting variables: tree size; age; form; species; condition; location; stand structure; site and environment conditions; and the presence and extent of tree defects. Trunk failures well above the ground line are relatively rare in hardwoods with high mechanical strength such as oaks and maples. This is not true for species with weaker wood and rapid decay, such as cottonwood, alder, and aspen. Moreover, once a potential tree is wounded, decay fungi may be confined to compartments within the tree through a process called compartmentalization, but the ability to compartmentalize varies by tree species. Over time, the tree may seal the wound with new wood resulting in a callus scar. The rate of wound sealing is a function of the tree growth rate and vigor. It is worth mentioning that many hardwood species, especially poplars, cottonwoods, maples, and alders, are more susceptible to branch failure than most conifers due to the hardwoods' lower resistance of branches to decay and the frequent occurrence of weak, narrow angles between branches. In contrast, most conifer branches are resinous and nearly at right angles from the boles. Additionally, long hardwood branches can be heavily weighted with foliage and fruit during the growing season. Early and late season wet snows, when foliage is present, also weigh down branches and tops causing breakage. Once damaged, poplars, cottonwoods, and birches are particularly likely to develop weaknesses and breakage because decay in these species can move outward and consume living sapwood, rather than being restricted to within a sound wood shell as in most other hardwoods [58].

## 5. Conclusions

The applied method of tree health condition assessment using acoustic tomography is an innovative research technique enabling non-invasive analysis of the internal tree trunk structures. Based on the analysis of the literature and the case study, it was possible to confirm the initial assumption that acoustic tomography may be an essential element in the sustainable management of urban greenery. This small case study of five trees contributes to the broadening of knowledge about this technique and its application in Poland. Thanks to the research presented, it was ordered that one poplar tree be left and observed in Lublin, that two linden trees be cut down in Sandomierz and Lublin, that arborist works consisting of lowering the height of a linden tree in Sandomierz be undertaken, and that monument protection be applied for an ash tree in Łęczna. In each case, the results of the tomography of the specimens confirmed the preliminary visual assessment of the trees.

This research verified the credibility of sound tomography for the promotion and wider use of this non-destructive method in tree diagnostics. In addition, the results of this research are expected to be used effectively to improve tree management in urban areas. Thanks to the tomography, trees classified as potentially threatening to safety are either observed or undergoing maintenance treatments correcting their statics. The correction of the tree silhouette is achieved by skillful cuts that reduce the weight of the crown or by mechanical reinforcements and supports, which does not remove the cause of the hazard, but protects the tree against falling for some time. The results of the diagnoses are presented on an ongoing basis, which means that we can assess the condition of the tree in the field. Presenting the test results together with a discussion of the operation of the tomograph, e.g., at press conferences, increases the residents' confidence in the authorities' decisions, e.g., to cut a tree. This principle also works the other way around where local community workers can, after presenting a dendrological report including a tomographic scan, find it easier to enter a tree into the register of monuments because officials accepting the application can be sure that the tree does not pose a threat to people and property. It is worth remembering that city residents should be educated and should decide on their own about shaping contemporary greenery in the city, and an essential element of a pro-environmental policy is social participation and environmental education.

**Author Contributions:** Conceptualization, M.D. and W.D.; methodology, M.D. and W.D.; software W.D., formal analysis, M.D. and W.D.; investigation, M.D. and W.D.; data curation, M.D. and W.D.; writing—original draft preparation M.D. and W.D.; writing—review and editing, M.D. and W.D.; visualization, M.D. and W.D.; supervision, M.D. and W.D. All authors have read and agreed to the published version of the manuscript.

**Funding:** This research received no external funding.

**Institutional Review Board Statement:** Not applicable.

**Informed Consent Statement:** Not applicable.

**Data Availability Statement:** Not applicable.

**Acknowledgments:** The research was financed from the own funds of the Department of Landscape Architecture and the Institute of Horticultural Production.

**Conflicts of Interest:** The authors declare no conflict of interest.

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
