# Peer review of "Sustainable Management of Very Large Trees with the Use of Acoustic Tomography"

_sustainability, doi:10.3390/su132112315_

Round 1

Reviewer 1 Report

The article presents an overview of large tree protection process in Poland, and how the use of the acoustic tomography can help to diagnose old tree specimens. 5 specimens were measured with this technique and the results were analyzed and contrasted with the opinions of experts in arboriculture.

As a general comment, I miss a comparison between the results obtained with the tomography and the opinions of the experts. It is said that these opinions were different, and I think that a paragraph should contain comments on these and which of them were more consistent with the results of the measurements.

L65. I would integrate this single heading in the General Introduction.

L68-69. What city are you referring to?

L74. "Branches and branches of trees ...". Consider revision

L85. "According to the Central Statistical Office (2018)". is it a reference?

L85-86. The largest group of what?

L87. Several categories? You only listed two.

L91-93. "The dimensions that qualify a tree or shrub to be considered a natural monument vary and —depending on the species - smaller in the case of shrubs and short-lived trees and larger in the case of naturally long-lived trees" . I don't understand this sentence

L111-112. Repeating and, and ...

L129-135. Same paragraph as L52-60

L137. Circumference at breast height?

L148. VTA. Not defined yet. Define it here.

L166. At what height were the waves measured?

L210-214. This paragraph should be in the introduction

L268. Is the limiting wall thickness defined in the Methodology?

L310. I don't understand the sentence

L341. In this case, there is no specific result for this tree. For example t / R etc. Please, be consistent in presenting the same results for each tree.

L403. Is this a result?

L414. Table 2 contains previous results (t / R). This result should appear before in another table, together with the percentages of affected wood.

L513. Data from the VTA assessments carried out by the experts should be provided.

Author Response

Dear Madam / Sir.

Thank You kindly for Your review and valuable comments.

We made all the corrections recommended. We enclose a new version of the manuscript.

The changes are visible in the text.

Yours faithfully,

Authors

Reviewer 2 Report

The article describes an experiment on the observations of tree health conditions that have been written well. Research on the application of sonic tomography technology on trees is not new, but this study was valuable for tree management.

There are some comments below to improve the contents of the article further:

Line 22: it is suggested to give brief information regarding the VTA results  (poor condition ??) as a basis for tree selection

Line 55: It is suggested to give a short explanation that the use of technology is the advanced level inspection after visual and basic inspection. You mentioned this explanation in conclusion line 517

Line 85-103: is there any information regarding tree health condition status for “monumental veteran trees”

Line 90: how large in “circumference at the breast height” of the tree which is becoming the reason to be protected as monumental tree

Line 129-133: is the statement repetition of line 55-59. It is not necessary

Line 136: what do you mean with “The work uses quantitative and qualitative data processing methods.”. It is not clear.

Line 142:”…. growing in selected cities in eastern Poland “growing in selected cities” should be added with more explanation.

Line 142: when the inspection was carried out should be informed since you provided figure of the tree as shown in Figure 3, 5, 7, 11.

Line 142-146: it should be added the coordinates for each trees’ samples address

Line 194-198: the sentence needs to be corrected. How can you say that the tomography result in percentage shows the speed of sound?

Line 217: Table 1 is suggested adding the columns for coordinates and VTA (results as mentioned in Line 148). Although you have explained the examination reason for each tree in line 220, 247, 305, 338, 368. It will be easy for to reader read the reason in the Table

Line 404-405: please make a clear sentence for the relationship of chemical properties in the result of observation by sonic tomography

Line 452: is it necessary to mention “the brand” of ARBOTOM as well as RESISTOGRAPH in this study

The conclusion is too long. You are suggested to focus only on the result and relate it to sustainability issues, as mentioned in the title.

Author Response

(The authors gave the same response as above.)

Reviewer 3 Report

The manuscript is well-written and fits well the general scope of Sustainability. The topic of the study presented is interesting and it includes a novelty. But, in current submitted form the manuscript has some minor shortcomings: Firtsly, both section Introduction and section Discussion must be added by new relevant references, which are highly cited on Web of Science and deal with the tomograph as a decision tool for very large tree management. Secondly, Authors should consider a terminology - I my opinion, the more frequent term is "very large trees" in literature (not "monumental trees"), "acoustic tomography or sonic tomography" etc. Eventually, Authors must add an international importance (and portability) of original results of the study to sections Abstract and Conclusion.  

Author Response

(The authors gave the same response as above.)
